# Environmental Qualities That Enhance Outdoor Play in Community Playgrounds from the Perspective of Children with and without Disabilities: A Scoping Review

**DOI:** 10.3390/ijerph20031763

**Published:** 2023-01-18

**Authors:** Thomas Morgenthaler, Christina Schulze, Duncan Pentland, Helen Lynch

**Affiliations:** 1Department Health, Institute of Occupational Therapy, Zurich University of Applied Science, 8401 Winterthur, Switzerland; 2Department of Occupational Science & Occupational Therapy, University College Cork, T12 AK54 Cork, Ireland; 3Division of Occupational Therapy & Arts Therapies, Queen Margret University Edinburgh Scotland, Musselburgh EH21 6UU, UK

**Keywords:** playthings, play value, affordances, inclusion, playground, outdoor play, environment, participation, vulnerable populations

## Abstract

For children, playgrounds are important environments. However, children’s perspectives are often not acknowledged in playground provision, design, and evaluation. This scoping review aimed to summarize the users’ (children with and without disabilities) perspectives on environmental qualities that enhance their play experiences in community playgrounds. Published peer-reviewed studies were systematically searched in seven databases from disciplines of architecture, education, health, and social sciences; 2905 studies were screened, and the last search was performed in January 2023. Included studies (*N* = 51) were charted, and a qualitative content analysis was conducted. Five themes were formed which provided insights into how both physical and social environmental qualities combined provide for maximum play value in outdoor play experiences. These multifaceted play experiences included the desire for fun, challenge, and intense play, the wish to self-direct play, and the value of playing alone as well as with known people and animals. Fundamentally, children wished for playgrounds to be children’s places that were welcoming, safe, and aesthetically pleasing. The results are discussed in respect to social, physical, and atmospheric environmental affordances and the adult’s role in playground provision. This scoping review represents the valuable insights of children regardless of abilities and informs about how to maximise outdoor play experiences for all children.

## 1. Introduction

Play is a fundamental right of children that is essential for health, well-being, and development, as stated in article 31 of the United Nations Convention on the Rights of the Child (UNCRC) [1]. The general comment 17 (GC 17) emphasises every child’s right to play and defines play as “any behaviour, activity or process initiated, controlled and structured by children themselves; it takes place whenever and wherever opportunities arise… play itself is non-compulsory, driven by intrinsic motivation and undertaken for its own sake, rather than as a means to an end” [2] (pp. 5–6). This scoping review examines children’s perspectives of play in playgrounds. Other research considers play from different perspectives, particularly its use to foster physical activity, or social and motor development [3,4,5,6]. This instrumental view of play has been critically discussed by scholars from multiple disciplines, including play-work, education, and health professionals, such as occupational therapists [7,8,9,10,11], who emphasise moving beyond such a perspective [11] and taking a reflective stance on how professionals value and utilize play through practice and research [8,10]. Instead, in line with the GC 17, professionals have begun to move to the consideration of how to provide time and space for play as an important approach to ensuring the right to play is addressed.

Playgrounds are one important environment mentioned in the GC 17 that should cater for play for all children, regardless of ability [1]. While playgrounds can exist in diverse community settings, for this review, they are defined as outdoor environments containing play opportunities provided for the purpose of play, located in public parks or schools available to the general public [12]. In many countries, community playgrounds are important spaces where children play [13,14,15,16,17] that are regularly visited by children and families of various ages and abilities [17,18,19,20]. Physical and social environmental qualities shape how, and in what outdoor play children engage [21,22,23]. Playgrounds have physical environmental qualities that are natural, built environments consisting of objects and spaces provided for play, and social environmental qualities that encompass potential opportunities to engage with others as well as attitudes, rules, and so forth [21,22,24].

As children are the main users of playgrounds, understanding how they use playgrounds, and what their wishes and preferences are for playgrounds should be the lynchpin of playground provision, design, and evaluation. By considering children’s perspectives on playgrounds, this study follows Rasmussen’s [25] differentiation between places for children compared with children’s places. Places for children can be playgrounds that are designed and built with adults’ ideas of what a playground should contain. However, not every place built for the purpose of play is a children’s place. Places for children can become children’s places when children connect to the playground through outdoor play, allowing them to attribute meaning to the playground environment [25]. 

Considering children’s perspectives has been found to contribute to positive outcomes such as meeting their needs [26], fostering community belonging and interest in spaces [27], and making spaces more inclusive [28,29]. Similarly, research has suggested that children’s perspectives as user-based knowledge are valuable and should include a broader range of diverse groups of children with and without disabilities in playground provision [13,23,28,30]. Despite the importance of considering children’s perspectives, previous research found that playground designers, planners, and providers have insufficient knowledge and experience in providing playgrounds that support play experiences for a diverse population, including children with and without disabilities [23,28,29,31,32,33,34,35,36]. Recent reviews [23] found that evidence relies on caregivers’ perspectives, such as parents, as a proxy for children with disabilities, and a “user-based knowledge including the broad range of diverse groups of children who are identified to be most at risk for play deprivation” (p. 17) should be considered in future research. Moreover, research from the United Kingdom and Switzerland concluded that good play provision needs to consider perspectives of children with disabilities and their families as an important reference point for other stakeholders [28,29]. Until now the perspective of children with disabilities in playground provision is often neglected [23]. Since playgrounds are places used by a diversity of children, this scoping review will include perspectives of children with and without disabilities to better understand what play experiences are important for such a diverse group of users.

To summarise, diverse children’s perspectives have not yet been sufficiently acknowledged in playground provision, design, and evaluation. Since playgrounds are built for the purpose of children’s play, children’s perspectives on playgrounds need to be taken more seriously and acted upon. This requires a better understanding of what children with and without disabilities seek in playgrounds. No other review has been found [13,14,21,23,37,38,39,40,41] that investigated published peer-reviewed literature that considered perspectives of children with and without disabilities and their play experiences in playgrounds. This scoping review aims to summarize the users’ (children with and without disabilities) play experiences and gain insight into what environmental qualities maximize the play experience in community playgrounds for all children. Such new synthesized knowledge will provide evidence considering children as informants and insight into how playgrounds can be understood as children’s places, where meaningful play experiences can take place.

## 2. Materials and Methods

A scoping review methodology was applied following the five stages proposed by Arksey and O’Malley [42]. Scoping reviews aim to systematically identify and map the extent, range, and nature of available evidence on a broad topic [43,44]. Scoping reviews have the advantage of allowing an extensive investigation of the entire scope of relevant primary research from a variety of disciplines regardless of study design and methodological quality [42,44]. Levac et al. [45] and guidelines from the Joanna Briggs Institute [44] were followed to ensure methodological rigor. A protocol was published prior to the investigation [46].

### 2.1. Identify the Research Question (Stage 1)

The following research question was formulated: What is known about how environmental qualities of public playgrounds contribute to the user experience of outdoor play among children with and without disabilities? We aimed first to summarize users’ (children with and without disabilities) experiences to gain insight into what environmental qualities maximize the play experience in public playgrounds for all children.

### 2.2. Identifying Relevant Studies (Stage 2)

A three-step search strategy was undertaken. First, an initial search helped identify relevant articles and other key search terms. Second, a test search in one database with an initial search string was performed. In this phase, information specialists from the University College Cork library were consulted to validate the search string and strategy. Third, the systematic search using the revised search string was conducted in August 2021; a follow-up-search was performed January 2023. See Table 1 for search terms. 

The systematic search of peer-reviewed studies was conducted in seven databases (Academic Search Complete, Avery Index to Architectural Periodicals, CINAHL, MEDLINE, PsycINFO, Scopus, and Web of Science) relevant to the disciplines of health, education, social sciences, and architecture, which allowed a broad range of literature for different professional audiences to be included. No limitations were set for publication year. The executed search string (see Table 1) was built using the Boolean operator OR between synonyms and the Boolean AND operator between concepts (playground, environmental qualities, and population). The search terms were applied to the text fields of title, abstract, and keywords in the included databases. Additionally, the Journal of Children and Youth Environments was hand-searched as it is an important journal for the review topic (February 2022).

### 2.3. Study Selection (Stage 3)

The two-step review process involved scanning titles, abstracts, and full texts. All citations were transferred into the online review software COVIDENCE [47]. Duplicates were removed. As recommended for scoping reviews, we applied an iterative review process resulting in the creation of inclusion and exclusion criteria. Title and abstract screening was divided into two rounds, with discussion of inclusion and exclusion criteria between the two screening phases. Round one helped identify the scope of studies and facilitated the identification of major areas irrelevant for this review, such as environmental hazards, safety concerns on playgrounds, or research on instrumental views on outdoor play such as play for physical activity, social behaviour, learning, health, development, or cognitive gain. Each paper was screened by two independent reviewers, and conflicts were resolved by a third independent reviewer. Studies considered questionable for inclusion were taken into the next round. Round two of the title and abstract screening was performed with the refined inclusion criteria found in Table 2. Again, each paper was reviewed by two independent reviewers, and disagreements between the reviewers were resolved in a group discussion within the review team. For the full-text review, each paper was screened by two independent reviewers following the inclusion criteria. Inconsistent decisions about papers were resolved in a team discussion. Reasons for exclusion were recorded in the full-text screening phase. To obtain an in-depth understanding of the published research scope, we first included perspectives of children and family caregivers. Studies that only represented family caregiver perspectives were excluded in the full text review (see Appendix A), but studies that contributed both the perspectives of children and family caregivers remained in this scoping review. However, only the children’s perspectives were included in further analysis.

### 2.4. Charting and Analysing the Data (Stage 4)

As recommended by Levac et al. [45], two phases of data extraction and analysis were completed. The first phase (by author T.M.) included the extraction of study characteristics into a Microsoft Excel spreadsheet. The data extraction form was piloted with three studies and approved by the review team to ensure relevance and clarity of extracted characteristics [44]. This data was analysed descriptively and provided an overview of the published research on the topic under investigation. 

The second phase included a convergent synthesis design using the same content analysis on qualitative, quantitative, and mixed-method studies [48]. The qualitative content analysis was guided by Graneheim and Lundman [49], starting with familiarisation with the data set by reading through all findings/results sections and note-taking relevant to the research question, as well as key study findings. The coding process started with a close reading of each study to first identify meaning units, which were formulated into condensed meaning units and abstracted to a code relevant to the research question. Multiple coding was applied if several meanings were identified in one meaning unit. The coding phase was subjective to the researcher’s (T.M.) interpretation in searching for patterns of meaning within the individual studies and throughout the whole data set. The coding process was performed in Atlas.ti 22 Windows [50] in two rounds. The codes were sorted into categories that were compared with each other and translated from one study to the next. In this phase, codes were refined. In the last phase, the overarching themes were formed through the identification of underlying meanings relevant to the research question of environmental qualities contributing to play experiences. Analysis started with a joint coding of the first four studies (approximately 8% of all studies) with the review team. The other studies were analysed by the first author (T.M.) and guided by discussions with the whole review team to confirm the formed codes and themes. The authors of this review are experienced occupational therapists, and two of them (C.S., H.L.) have advanced knowledge of research on children’s play. 

### 2.5. Collecting, Summarizing, and Reporting the Results (Stage 5)

The findings where charted and summarized by the first author (T.M.). First, the scope of the included studies is summarised. In the second part, themes from the qualitative content analysis are presented. In approaching this review from a children’s perspective, including children with and without disabilities, the findings only elaborate on a specific population when findings were only found for that specific population.

## 3. Results

The searches revealed a total of 6095 references, which was reduced to 3190 after duplicates were removed. Articles were reduced in two rounds of title and abstract scans first to 503 studies and, after refinement of the inclusion criteria, to 104 studies. Full text reviews identified 49 studies meeting the inclusion criteria (see Table 2). Two additional papers were identified in a hand search resulting in the final sample of 51 studies. For the selection process, see Figure 1.

### 3.1. Descriptive Numerical Analysis of Included Studies

A detailed description of the included papers can be found in Appendix B. The publication dates range from 1974 to 2022, (see Figure 2). Most studies were conducted in Europe (*n* = 25), Australia and New Zealand (*n* = 11), followed by Asia (*n* = 8) and North America (*n* = 7). No studies from Africa or South America met the inclusion criteria (See Figure 2).

Interdisciplinary contributions to the topic came from Architecture and Landscape Architecture (*n* = 16), Education (*n* = 15), Occupational Therapy (*n* = 6), Psychology (*n* = 4), Exercise, Sport and Nutrition Sciences (*n* = 4), Human Geography (*n* = 3), Occupational Science (*n* = 1), Public Health (*n* = 1), and Practitioner Researcher (*n* = 1). 

Most papers employed qualitative (*n* = 30) methods, followed by quantitative (*n* = 13) and mixed methods (*n* = 8). Most studies used multiple data collection methods (see Table 3), and the most common methods were semi-structured interviews (*n* = 23), focus groups (*n* = 10), walk-along talks (*n* = 10), and observational methods referred to as systematic (*n* = 11) and unsystematic (*n* = 9).

**Figure 2 ijerph-20-01763-f002:**
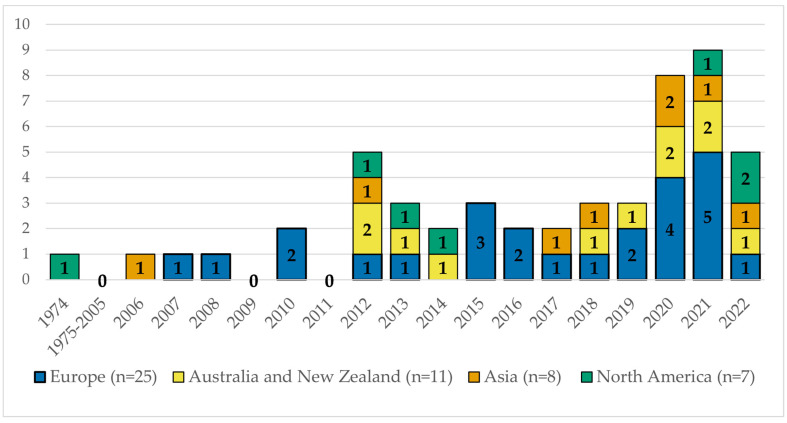
Number of included publications in the review period 1974–2022 (*n* = 51) by continent.

#### Playground and Participant Characteristics

Several reviewed studies shared the same data set. These were four studies from Sweden [56,57,58,59], two studies from Norway [97,98], two studies from Australia [75,76] and two studies from Switzerland [72,74]. These papers have only been included once in the following description of participants and playground characteristics.

In total, 212 playgrounds were represented. The locations of playgrounds were reported in urban (*n* = 19), suburban (*n* = 6), rural (*n* = 6), and mixed (*n* = 2) areas. Location was missing in ten studies, and in two studies a reported location was not relevant to the study aims. Playgrounds were located in community areas such as playgrounds in parks or specific playground spaces (*n* = 27), schools (*n* = 16), or both public and school playgrounds (*n* = 3). Two of the papers looked at adventure playgrounds [54,81], and two studies focused on inclusive playgrounds [72,95]. 

The total participant sample included *n* = 3676 children with 34.9% (*n* = 1282) male, 35.3% (*n* = 1299) female, and 29.8% (*n* = 1095) unreported sex. Six observational studies provided observation counts but not participant numbers [51,89,93,94,95,101], and one study did not report participant numbers or sex [66]. The study population age range fell most frequently between 5 and 10 years (31 studies), with 20 studies including children aged 11–12 years and 21 studies including those aged 3–4 years. The least represented age groups were the youngest and oldest populations: 0–2 years (7 studies) and 13 and older (6 studies).

Only ten studies included children with disabilities, representing a sample of 125 participants (approximately 4% of the total sample). Of these, four studies focused on children with disabilities only [53,60,68,102], and six studies included mixed participant groups of children with and without disabilities [61,67,72,74,77,95]. The following disabilities were represented in the studies: motor-related disabilities [60,61,67,72,74,77,95], autism spectrum disorder [53,60,61,72,77], visual impairments [60,61,67,72], developmental disabilities [60,61,67,72], hearing disability [60,72], intellectual disability [72,77], and learning disability [60]. 

Fifteen studies also included other participants along with our population of interest, for example, parents and other family members, teachers, playground maintenance staff, play workers, or health professionals [51,52,53,56,58,59,60,61,64,66,68,83,88,99,102]. Findings from these participant groups were not included in the qualitative content analysis.

### 3.2. Findings from Qualitative Content Analysis

Five themes were identified that provide insight into the interconnection between play experiences and the environments that enhance outdoor play (see Table 4). These five themes describe all children’s preferences and desires for play experiences and their relation to the environment. The themes are described with an explanation of how the play experiences relate to the social and physical environment (see Table 4). The themes are presented individually but are not distinct from one another, since play experiences are not mutually exclusive and are experienced simultaneously and in combination through outdoor play.

#### 3.2.1. We Seek More Intense Play Experiences

Seeking intense play experiences describes all children’s desires for **diverse intense and novel play experiences,** which related to engagement in intensified movements and intensified sensory experiences (see Table 5). Across studies, children consistently sought more intense play experiences, relating to faster, slower, longer, heavier, deeper, bumpier, curvier, further, and more elevated play. These intense play opportunities were directly linked to the physical environment and were captured by children’s emphasis on bigger, taller, higher, wavier, and longer play equipment landscape features, and objects [51,53,61,66,69,70,73,84,90,92,96,100]. Table 5 describes intense play experiences and environmental affordances with supporting studies. Intense play experiences were relative to the child, meaning desired intense experience differed from child to child. For example, great speed was sometimes described as causing feelings of dizziness or nausea [57,64,70], in some cases leading to children actively avoiding such play equipment [77]. However, a further six studies described how children combined intense play experience and sought more intense play [56,57,63,69,73,88]. Additionally, an indication of a diverse provision of intense play experiences is given in children’s emphasis that only one intense play opportunity was insufficient [51,58,62,70,78]. Children also compared playgrounds with one another and expressed how the novel qualities of one unique playground (see Table 5) contributed to the experience of intense play.

#### 3.2.2. We Want to Make Our Own Choices about What to Play

This theme described children’s wishes to choose and self-direct their play, identifying three aspects of how children engage with their playground environment: (1) finding suitable challenges, (2) using the environment flexibly, and (3) having moments to unfold their own play. This theme synthesises evidence of what the physical and social environment provided for children, especially in the context of having permission to self-direct their own play.

Children reported the importance of having **suitable challenges** in their playgrounds [59,61,78,84,88]. Suitable challenges meant having the choice to gradually engage in more challenging play. A connection with intense play affordances was identified. Suitable challenges should not be too easy [53] and needed to match children’s ages or abilities [61,84]. This was reported when a range of the same play possibilities with different difficulty levels was provided [54,61,63] that allowed children to choose a difficulty level according to their abilities [61,68,77,94,102]. Similarly, for children with disabilities, a range of difficulty levels from easy to advanced play opportunities needed to be provided [61,68,77,102]. For example, diversity in provision allowed choice of where to climb up/down, or to use a ramp instead of climbing-ropes [77]. Children with disabilities expressed concerns and sadness when no suitable challenges were available for them compared with those provided for peers without disabilities [61,67,68]. In such cases, children with disabilities needed to rely on their caregivers’ support, which meant they were not able to self-direct their play [61,67]. This made children feel alienated compared with their peers [61] or resulted in not using playgrounds at all [67].

**Using the environment flexibly** meant exploiting any possible way to engage with play equipment and other built structures beyond their intended use [53,54,57,58,60,61,64,65,66,67,71,72,77,80,84,87,88,90,92,101]. Flexible use showed that children utilized any available spaces and objects provided in their play, blurring the boundary between where a playground starts and ends. Examples of flexible use included socializing on play equipment through sitting and chatting and not being active [51,57,67,71,79,80], climbing up swing posts [58,59,65] or play houses [57,67,88], balancing on rolling bars [88], using play equipment to hide [57,59], jumping over sand pits [84], or hanging from play equipment such as basketball baskets [84]. Other children exhausted affordances for play in a diversity of approaches to play equipment, such as sliding-play by sliding on their tummy, backwards, head-first, or climbing a slide [53,65,67]. Additional loose materials such as water, sand or stones on slides [54,71,89] or filling spinning equipment with surface materials [59] were applied to play equipment. Besides play equipment, every built structure was incorporated into play, but this was mainly elaborated on in studies that focused on children without disabilities. Flexible use of built structures like fencing afforded balancing, climbing, and jumping over [65,66,78,84,92], walls used for ball play [75], and any other low raised boundaries such as those surrounding trees used in socio-dramatic play [63,75,76]. Benches and other seating opportunities afforded climbing, balancing, and jumping from [51,63,75]; tables and seats were used to play house and supermarket [75,80]. Other built structures utilized for play were stairs, pillars, lampposts, window blinds, and exercise equipment intended for adults [53,63,65,75,78]. 

**Unfolding own play** was linked to social and physical environmental qualities. This was accomplished by having uninterrupted moments to unfold own play afforded by the physical and social environment. The physical environment meant having a suitable space and objects affording children to engage in their unfolding play. The social environment meant adults (and other people) granting permission for children’s own play. Across studies, children or children’s groups provided examples of their unfolding play in correspondence to their particular physical and social environment [53,55,56,57,61,63,66,67,68,70,75,76,81,88]. Unfolding own play evolved during play itself, while the child was engaging in own play within and with the environment such as climbing-high-and-performing-climbing-stunts-play [57,70], fly-jump-off-the-bench-with-closed-eyes-play [63], jumping-off-swings-and-rolling-down-with-peers-on-slope-play [73], needle-spotting-play [55], or race-against-the-rolling-disc-play [53]. Sometimes, unfolding play meant to invent one’s own rules [53,70,74,75,77,78,91,94]. Sometimes, unfolding own play was evident in imaginative play such as crocodile-pulls-me-down-the-slide-play [67], being-a-superhero-jumping-off-the-bench-play [63], the-floor-is-lava-play, spinning-object-portal-play [53], being-dragons-protecting-the-old-tree-play, or cooking-as-mum-and-dad-play within the protection of low-hanging-branches [76]. These examples illustrated that children unfold their own play in correspondence within and with their social and physical environment. 

#### 3.2.3. We Value Both Playing with and away from Children and Adults

This subtheme describes the evidence showing that children value and seek opportunities to engage with a diversity of people and animals while also wanting to play by themselves, especially when playing away from adults offers treasured play experiences. 

**Playing with children** was a key component described as enhancing children’s play experiences. Examples included swinging together with one particular friend [54,78], rolling down hills and feeling dizzy with peers [70], climbing with one friend on a tree [78,96], crashing into each other while swinging side by side [54,64], or having more time to talk and hang out with other children [55,67,68]. Children with and without disabilities described wanting to play with peers with similar interests, of the same sex, and of similar age and ability, as this offered engagement in playful competitions and belonging [53,54,70,72,74,75,76,79,84,88,91]. This was found for varying social forms such as two children only, small and big groups, two equal-sized groups, or gendered groups [72,74,75,76,91,101]. Playing with other children meant not always doing exactly the same things, but still being included; for example, swing play meant pushing the swing while cheering friends sat on the swing [77]. Similarly, climbing meant doing the same activity differently, such as a child in a wheelchair climbing at ground level and peers climbing on the same structure but higher up [72]. The physical environment afforded possibilities to connect with other children while playing [51,60,67,68]. These environmental features included multi-player equipment that accommodated several children or allowed parallel play [51,58,60,67,72,79,84,95], play equipment placed in visual proximity such as circular and parallel placed swings [54,64,82], or small spaces that encouraged playing with or sitting and talking with peers [67,68,101]. This means playing with peers sometimes included playing nearby, following each other’s lead, or using other children as a source for play ideas [53,72,92,96].

Adults who accompanied children influenced their play [52,67,88]. **Playing with adults** occurred in the presence of supportive adults who knew how and when or when not to intercede with children’s play. This was perceived in three ways. First, adults were actively involved in the same play together with the children [52,64,84,88] such as helping in constructing play [81] or playing together on the same play equipment [52,68,71]. Second, sharing play, wherein adults watched the child rather than being active themselves [52]. This was shown through a child’s desire for their parents to witness how fast they could slide down, or when the child got approval from an adult to engage in a certain play occupation [54,73]. The availability of supportive adults who assisted in a difficult situation was important, as this enabled children to try new challenging play [60]. Third, adults and parents were not involved in the play and only accompanied the child to the playground [52,61,67,88]. This was particularly important because children indicated a strong wish for less adult surveillance [54,67,68,84] and opportunities to socialise with peers [67,68].

**Being away from adults and other children** was about the experience of not being seen and showed a connection with the theme of choices and self-directed play. Children referred to being away from adults more frequently than they did being away from other children. Being away from adults was associated with privacy, having secrets, and breaking the rules. This was possible in places in the natural and built environment that afforded privacy and seclusion, such as small spaces [59,64,67,75,76,81,96], in-between-spaces [64,66,75,76,78,94], and out-of-bound-spaces (places where children were not allowed) [56,64,75,76]. These places were out of the supervising adult’s gaze and allowed uninterrupted play [62,64,67,75] and resting and observing [94]. Other studies found that smaller fully- or partly-enclosed spaces elicited homey, cosy, and relaxed feelings [63,75,81], and sometimes small spaces in the natural environment were described as secret places nobody else knew about [63,64,81]. In addition, children not under direct adult surveillance started to test boundaries by breaking adult-created rules [67,84], such as climbing up to a high point or hanging upside down [67], climbing on play equipment or trees not meant for climbing [57,76,84], fast running [53,84], digging under play equipment [59], wandering off to out-of-bound-spaces [55,64,75] like a secret path in forestlands [63], or entering prohibited spaces [76]. There seemed to be a shared understanding between children that breaking such rules is an acceptable way to make play more exciting [57,66,67,76,84].

#### 3.2.4. We Want to Belong to Our Playgrounds

Both physical and social environmental qualities influenced children’s feelings of connection with their playgrounds. Belonging to their playground was experienced by (1) being familiar with the playground, (2) feeling welcome and safe, and (3) enjoying playground aesthetics.

**Feeling connected with the playground** had relational qualities, including knowing the playgrounds located in the home community [55,58,61,66] within walking distance and regularly visited [54,57,59,61,62,66,67,70,78], and by knowing the people using the playground [54,72,73,78]. Playgrounds provided a space to get together with friends and fostered potential social opportunities, including making new friends [52,67,74,77,81]. The connection was strengthened by knowing that playgrounds were distinct children’s places where children are the primary users [61,62,66,79]. However, four studies pointed out that playgrounds were also family spaces [52,55,61,68]. Children connected with their playgrounds when opportunities to shape their environments were given; that is, building their own dens or leaving permanent markings on the physical environment [64,81,96].

Children wanted to **feel welcome and safe** in playgrounds, especially children with disabilities who wanted a welcoming atmosphere created by accessible and usable playground design that afforded play with others [60,61,67,68]. This was possible when children had opportunities to participate in identifying changes to make a playground playable for children with disabilities [60,67,72,102]. Welcoming feelings were undermined by attitudinal and othering practices such as name-calling, refusing to include children in play, or staring [60,68,72]. Regardless of ability, children expressed the importance of feeling safe because of the availability of others, including known friends [62,80], people in the community [55], caring adults [60,64,80], or when past positive experiences were associated with a playground [61]. Gang activity and hazardous litter contributed to feelings of danger [55]. Similarly, feeling unsafe was reported when dangerous traffic was nearby [62,80] or when secluded spaces fully separated children from others in the place [75]. Whether fences contribute to a feeling of safety was unclear. Fences functioned as a boundary from dangerous situations [64] but also limited play possibilities [76] or gave children a caged feeling [84]. However, three studies found that children felt safer with fences [70,83,85]. 

The connection to playgrounds was strengthened by children’s **perceptions of aesthetics and beauty** in the built and natural environment. Children wanted to play on appealing playgrounds. Playgrounds were unattractive and ugly when they were dirty; smelled bad; were littered with glass, needles, cigarette butts, duck and dog excrement; had graffiti; or were noisy and overcrowded [53,55,63,64,78,84,92]. A lack of bins and other amenities that would be useful at playgrounds, including toilets, water fountains, and changing rooms, was noted [60,70,71,78,82,89]. Similarly, five studies pointed to mouldy and rotten, broken, and damaged play equipment [56,57,67,80,84]. Four studies found that the natural surrounding space was overgrown, hindering children’s access to play [70,75,76,89]. Beautiful and appealing playgrounds were related to a naturalized and colourful provision, and a wish for more colour, including colourful flowers, leaves, plants, playground equipment, and buildings was reported [63,64,65,71,76,81,84,85]. The importance of natural provision was expressed in the appeal of more natural elements, such as gardens, trees, flowers, fruits, grass, bushes, rocks and boulders, water and a variety of natural loose materials [63,64,70,71,76,79,82,85,89,91,94]. Nature provides additional play possibilities [70,78,84], including exploration and discovery [57,58,64,78,81], connection with animals [55,64,91] and sensory play (smelling, tasting, touching, rubbing, observing,) [68,76,78,79,88,94]. A more natural environment contributes to a more relaxing and restful atmosphere [55,63,70,75,76] and helps children feel calm [63,81]. 

#### 3.2.5. We Desire Fun

The **experience of fun** was the overarching and core experience in outdoor play. Children actively sought more fun in play. Fun evolved while playing but was also anticipated when children engaged in play and was sometimes described as a prerequisite for play [57,61,84]. Children reported that an activity was play when it was fun [84], and it was not play when fun was lacking [57]. The experience of fun was essential and interwoven in children’s outdoor play, a common thread that connected to other themes. Both physical and social environmental qualities were associated with creating fun play experiences.

For the physical environment, this was interwoven with providing intense play opportunities [54,57,59,84] and suitable challenges allowing for success and the experience of mastery and achievement [53,67,81]. Children described fun as the feeling of a little bit of danger such as hanging upside down [57,67,81], play that made children dizzy [57,64,70], or even getting a little bit hurt, such as when jumping down from elevated heights [57]. Fun was associated with certain popular playgrounds that were visited more often [58,70], playgrounds were compared with each other to elaborate on what was fun on a particular playground [57], and children had ideas on how to keep playgrounds fun for longer periods of time [64,70,84].

The social environment included having people and animals available to play with [54,55,61,81], which was associated with having something to do [53,54,55,61,70,84] and experiencing new memories together [52,61]. In contrast, having nobody to play with was not fun [67,77]. Another social aspect was having permission from adults and other children to engage in fun play [53,55,66,76,84]. Fun meant allowing children to self-direct play [53,67], engage in play in a unique way that matched their abilities [59,72,77], or being allowed to engage in new challenges [67].

Fun was contrasted to the boredom experienced when the playground did not provide a sufficient diversity of play opportunities in the built and natural environment [54,55,57,58,64,70,79,84]. This was also associated with repetition, and doing the same thing repeatedly led to the feeling of being fed up [78,84]. Additionally, existing playgrounds did not present enough challenges [59,61,78,84,88] for older children [57,61] and suitable challenges for children with disabilities [67,77]. This meant a fun playground needs to provide both suitable challenges with ways for evolving complexity that caters for all children regardless of their ability or age [61].

## 4. Discussion

This scoping review aimed to summarize the experiences of children with and without disabilities, and to gain insight into environmental qualities that maximize play experiences for all children on community playgrounds. There were two key findings. First, this review showed that the combined qualities of the physical and social environment afforded play experiences that children preferred and desired when engaging in outdoor play. This interconnection was sometimes linked to certain environmental qualities relating only to the physical environment or only to the social environment, but more commonly related to both the social and physical environment combined. Second, the review revealed children were knowledgeable about their community playgrounds and environmental qualities that contributed to enhancing their outdoor play experiences. These play experiences were multifaceted and included having opportunities for fun and intense motor and sensory play, engaging in suitable challenges, making choices, and having moments to unfold their own play. Children valued a spectrum of play, from playing alone, to playing in small to big groups of peers and friends, and playing with adults and animals. Children also desired safe, welcoming, and aesthetically pleasing playgrounds where they felt they belonged as they knew other users and the playground. In other words, community playgrounds with the best play value become a *children’s place*. These findings point to the importance of acknowledging children’s perspective’s, regardless of ability, in playground provision, design, and evaluation.

### 4.1. Physical, Social and Atmospheric Affordances for Outdoor Play

This review presented rich and varied information about how children utilised physical and social environmental affordances available to them. Environmental affordances have been successfully used in outdoor play research to understand how children perceive and use their environments for outdoor play [64,69,76,79,87,90,92,96,97,103,104]. This idea of affordances draws from the concept originally coined by Gibson, who describes affordances as possibilities for action that are perceived and actualized by the child in relation to the environment [103,104,105]. The findings in this review presented a variety of affordances in relation to physical, social, and atmospheric environmental qualities that provide insights into play experiences for *children’s places* that went beyond simply being a place to play.

The findings of the review confirmed a persuading agency of the physical environment on affordances that children perceive and actualize in outdoor play. Persuading agency refers to the power the environment has to entice children to play. In this review, the understanding of affordances was broadened, as children did not only perceive action possibilities [92,103,104] such as sliding, running, or climbing. Rather, children regardless of ability also perceived the intensity, novelty, and challenge in potential action possibilities, such as swinging, rolling, or digging that was faster, slower, longer, heavier, deeper, bumpier, curvier, further, and more elevated (see Table 5). While these intense play affordances sound like the idea of risky play [106] the review findings showed more diversity and choice was associated with the concept of intense play affordance. Intense play affordances were not always about being risky or adventurous, but rather included a broader diversity in movements and sensory experiences, including doing something intentionally slower, experiencing intense tactile or auditory sensations, and combining intense play affordances for an even more intensified experience. These suggest the importance of providing a diversity of intense play experiences in playgrounds.

Intersections between the physical and social environmental qualities were found to contribute to the overall atmospheric and more tacit atmospheric affordances of playgrounds. Children elaborated on these atmospheric affordances in their experiences of feeling safe and welcomed and their perception of aesthetics. A physical environment that is aesthetic, colourful, clean, and contained both built and natural play opportunities contributes to an environment that is appealing to children. Loebach and Gilliland [17] found that children are well aware of atmospheric qualities afforded by their physical environment, such as recognising poor aesthetics and conditions. Other research has elaborated on the importance of the nature provision of playgrounds and its contribution to atmosphere [24,107,108,109]. Regarding natural environments, an interesting finding from this review was that studies exploring the outdoor play of children with disabilities merely focused on the built environment, such as play equipment or surfacing, whereas studies that explored perspectives of children without disabilities elaborated to a much greater extent on the natural environment as well as other built structures (such as benches, fences, stairs) for play. Other scoping reviews that investigated playgrounds and outdoor play of populations with disabilities corroborate this finding [13,23]. Yet natural environments provide potential affordances for sensory play for children with disabilities [41]. This points to the need for future research into how children with disabilities use natural environments for play in playgrounds [110,111] and how playgrounds can provide more nature access for all children, regardless of ability. Besides the lack of natural environments in the studies with populations of children with disabilities, the built environment, especially physical environmental qualities such as accessibility and usability in transaction with social environmental qualities such as attitudes, were strongly related to whether children with disabilities experienced a welcoming atmosphere. This finding is supported by previous research that only looked at children with disabilities and their caregivers’ perspectives [13,23,40,41] and exemplified that social and physical environmental qualities relate to atmospheric affordances.

In this review, social affordances were not only related to those with whom children played, such as children, adults, or animals. Children, regardless of ability, emphasised their relationship to significant play partners such as children and adults they knew, such as friends from school and the proximate neighbourhood, known people from the community, and known peers with similar abilities and interest, or of the same sex. This concept of knowing others also transcended to the physical space, as if the playground was a friend as well. Children wanted to feel connected to their playground through positive play experiences either alone or with known people who were associated with the playground. These findings indicate that playgrounds have the potential to be spaces for social inclusion [28,29]. Supporting social inclusion, therefore, needs to consider both built environments, such as inclusive design solutions, and involvement of the local community, including children, caregivers, and other stakeholders in respecting their needs and preferences for the playground [23,28,33]. Including local community perspectives, such as those of local children, can build a foundation in coming to know the people and the community alongside building a connection with the playground. This has the potential to nurture children’s sense of belonging to the space and their communities. 

Further social affordances in this review were related to social rules and practices such as having permission to use built environments flexibly, having uninterrupted moments to unfold own play, and having opportunities to play away from adults and solely with peers. Both children with and without disabilities valued opportunities to self-direct their play an expressed the wish for less adult interference and surveillance. However, such opportunities were identified differently by children with and without disabilities and depended on different physical and social environmental qualities in combination. For children with disabilities, adult permission needs to be discussed, since these children frequently reported needing adult assistance due to inaccessible and unusable physical environments [61,67,68,102,112]. Other studies from children’s, parents’, and professionals’ perspectives identify parent and caregiver support as a barrier to engaging with peers and self-directed play [67,113,114], which further makes children with disabilities feel alienated and embarrassed [61,112,113], and this dependency on adults is not perceived as fun [112]. Depending on adults to overcome barriers in the physical environment limits children’s social opportunities in engaging with other children on playgrounds [23]. For children without disabilities, such barriers in the physical environment were not an issue and, therefore, not represented in the literature reviewed. Consequently, to enable outdoor play for children regardless of ability, this scoping review’s findings suggest two elementary considerations. First, playgrounds need to be physically accessible and consider usability considering diversely-abled children. Here, attention needs to be given to providing equal play opportunities to all children, especially opportunities to play with other children. Second, adults need to know when and how they influence children’s play, and when they should step back and give children more permission to experience self-directed play.

### 4.2. Adult’s Role in Outdoor Play Provision

The review findings provide synthesized evidence of children’s perspectives on the outdoor play experiences they value and prefer in community playgrounds, helping to identify environmental qualities that provide for such experiences. Aligning with other studies [13,23,26,29,30,33,72], this scoping review showed that children with and without disabilities are knowledgeable users of playgrounds who need to inform playground provision, design, and evaluation. Considering children’s perspectives reflects Article 12 of the UNCRC [1] on children’s right to be heard in matters that affect them while giving them “due weight in accordance with the age and maturity.” As stated in the GC17, all children should have a central role in playground provision [2]. Therefore, navigating all children’s rights to be heard in matters such as playground provision needs a clear standpoint on the role of adults in playground provision. All children can form and express their views, but it is the adult’s responsibility to facilitate this participatory process [115,116,117]. This role requires adults to be informed on children’s perspectives and how to incorporate these perspectives into the design of physical and social environments that maximise play experiences. Therefore, an adult’s role in play provision, design, and evaluation is to be an adult ally of children, which means being informed about and supporting children’s perspectives as well as serving as “bridging persons” [116] (p. 342) between children and adult stakeholder perspectives. The scoping review findings inform adults about environmental qualities that enhance children’s play experiences. Certainly, designing children’s play spaces means an adult perspective should not overshadow what children value in their playgrounds and outdoor play [25].

### 4.3. Strengths and Limitations

The findings need to be interpreted with the following considerations. First, parents’ and caregivers’ perspectives were not included. These would provide an additional viewpoint for understanding children’s play experiences. Future research might investigate both the perspectives of parents and children to provide insight into how these differ. Second, the included studies encompassed community and school playgrounds as the units of analysis. This inclusion criterion was set because, for some countries, school playgrounds are open to the public during non-school-hours. In this review, most studies of school playgrounds did not specifically state if a school playground was open to the public or not. However, some included study data were collected during school times. Third, a methodological quality assessment might strengthen the interpretation of the study findings. However, a methodological quality assessment was omitted due to the interdisciplinary scope of the research and the variety of methodologies and methods used. A scoping review was considered a suitable methodology in the interdisciplinary area of playground research, and the inclusion of publications from a diversity of disciplines strengthened the findings. 

### 4.4. Future Research

This review was able to link play experiences to environmental qualities by synthesizing findings from peer-reviewed articles using qualitative, quantitative, and mixed methods. The most useful information regarding children’s perspectives was collected through multiple methods, including verbal accounts (such as go-along interviews) and observations. Most of the studies relied on qualitative methodologies. This highlights the need for instruments to allow investigations into children’s perspectives in a more systematic way and consider the conjunction between children’s experiences and environments. Future research should investigate environmental qualities such as those found in this scoping review to elicit potential affordances for play experiences available to children with and without disabilities. 

A further gap in research was identified in the representation of perspectives in research. Only ten studies included children with disabilities; of these, only six studies included children with and without disabilities in combination. If playgrounds are places for inclusion, research and practice need to take a more diverse perspectives into consideration in playground provision, design, and evaluation.

## 5. Conclusions

This scoping review aimed to synthesise play experiences of children with and without disabilities and gain insight into what environmental qualities contribute most to enhancing play experiences in community playgrounds. Gaps in research were identified in the limited number of papers that included both children with and without disabilities, in the lack of research about how community playgrounds can provide more access to nature for all children, regardless of ability, and the need for instruments that investigate the connection between children’s experiences and environments. The main findings of this scoping review were as follows. First, the available evidence allows an understanding of how the combined social and physical environmental qualities of playgrounds enhance outdoor play experiences by providing a diversity of experiences. Multifaceted play experiences were reported with the desire for fun, challenging, and intense play; the wish for self-directed play; opportunities to play alone and with known social partners; and a desire for welcoming, safe, and aesthetically pleasing playgrounds. Second, regardless of ability, children were knowledgeable about the play value of their community playgrounds, and therefore, their perspectives need to be more closely considered. Playground provision, design, and evaluation needs to move beyond merely providing dedicated spaces for play and consider instead, provisions for potential outdoor play experiences that allow playgrounds to become children’s places. This means that children’s preferences and what children want to experience in playgrounds need to be at the heart of playground provision, design, and evaluation.

## Figures and Tables

**Figure 1 ijerph-20-01763-f001:**
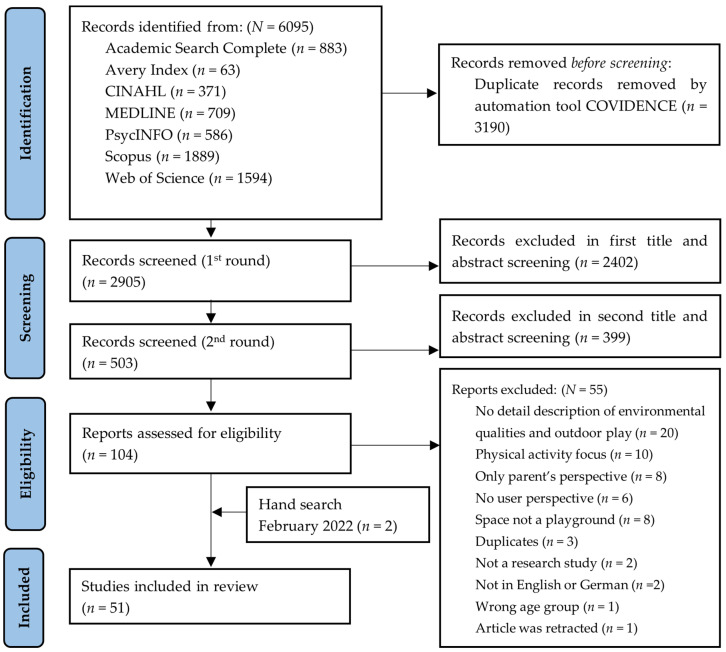
Prisma flowchart identification, screening, eligibility, and inclusion of studies.

**Table 1 ijerph-20-01763-t001:** Search terms.

Concepts	Search Terms
Playground	playground * OR playscap * OR playspac * OR “play spac *” OR “play area *”
Environment	buil * OR design * OR provi * OR natur * OR outdoor OR inclusive
Population	child * OR kid * OR caregiver * OR parent * OR mother * OR father * OR famil *

Notes: *** asterisk was used as truncation command in databases search.

**Table 2 ijerph-20-01763-t002:** Inclusion and exclusion criteria.

Inclusion Criteria	Exclusion Criteria
The primary focus is on public playgrounds built for the purpose of play, including playgrounds in community environments such as parks and schools.	Spaces used for play but not built for the purpose of play (e.g., forest areas used for play, studies concerning community neighbourhoods)
Studies that consider environmental qualities and outdoor play	Studies that only investigate environmental qualities (e.g., accessibility) and do not relate environmental qualities to outdoor play
Studies about outdoor play considering play for the sake of play	Studies about outdoor play for other means: (1) health reasons (e.g., outdoor play for physical activity), (2) learning, (3) restoration, (4) social interaction, or (5) interventions
Studies with populations of children and youth between 0 and 12 years, with and/or without disability.	Studies only addressing children and youth populations older than 12 years of age. Studies addressing only family caregiver (parents, legal guardians) and expert perspectives (e.g., teachers, landscape architects, playground providers, health professionals)
A primary peer-reviewed study including qualitative, quantitative, and mixed methods	Not a primary research study (e.g., opinion pieces, editorials, systematic reviews, methodological papers, historical papers, papers concerning measurement development, conference proceedings)
Written in English or German	Languages other than English or German

**Table 3 ijerph-20-01763-t003:** Data collection methods of the included publications.

Methods	Study Reference Number	*n*
Verbal accounts		
Semi-structured interviews	[51,52,53,54,55,56,57,58,59,60,61,62,63,64,65,66,67,68,69,70,71,72,73]	23
Secondary analysis of semi structured interviews	[74]	1
Focus groups	[56,57,58,59,75,76,77,78,79,80]	10
Walk-along talks	[55,61,63,64,70,75,76,78,80,81]	10
Play along	[69]	1
Creative methods		
Children’s drawings	[53,63,71,79,80,82]	6
Playground model creation	[63,79,80,81]	4
Picture sorting	[53,83]	2
Scrapbooking	[77]	1
Children taking pictures	[63,77,80]	3
Referred to participatory methods	[79,80,84,85]	3
Referred to Mosaic approach [86]	[53,63,77,85]	4
Observational methods		
Systematic observation (including behavior mapping)	[51,54,62,73,75,76,87,88,89,90,91]	11
Unsystematic observation	[52,53,58,65,66,69,72,77,92]	9
Observation using instruments	[83,91,93,94,95]	5
Systematic video observation	[96,97,98]	3
Unsystematic video observation	[92]	1
Taking photographs (by researcher)	[51,52,53,56,58,59,62,63,65,69,75,76,78,81,89,90,92,96]	18
Auditing playgrounds	[61,91]	2
Field notes	[51,52,53,56,62,66,69,72,78,81,87,92,93,96]	14
Questionnaires/Surveys	[52,55,56,65,88,99,100]	7

**Table 4 ijerph-20-01763-t004:** Enhancing outdoor play: The relation between children’s play experiences and the physical and social environment.

Theme	Play Experience	Relation to the Physical Environment	Relation to the Social Environment
We seek more intense play experiences	Intense play including faster and slower, longer, heavier, deeper, bumpier, curvier further, and more elevated play	Provide a great diversity of intense and novel play opportunities.	
We want to make our own choices about what to play	Finding suitable challenges	Provide a range from easy to complex play opportunities.	
	Using the environment flexibly		Permission to exploit play opportunities of the provided physical environment.
	Having moments to unfold own play	Uninterrupted moments for play (having the space).	Uninterrupted moments for play (having adult permission).
We value both playing with and away from children and adults	Playing with children	Consider social-physical qualities such as multi-player equipment, proximity placement of play equipment, and space qualities such as small spaces.	Availability of peers and friends that children can relate to.
	Playing with adults		Supportive adults that know when to play with the child and when not to intercede.
	Being away from children and adults	Social-physical qualities such as a variety of spaces allowing not to be seen.	Understandable adult-created rules. Possibilities to flee adult surveillance.
We want to belong to our playgrounds	Feeling connected with the playground	Location of playgrounds in own community.	Knowing the space and people, having social opportunities.
	Feeling welcome and safe	An accessible, usable, well-maintained playground with fences to protect against dangerous situations.	No othering practices. Opportunities to participate in changes made to the playground. Knowing the people.
	An aesthetic and beautiful playground	Colorful provision. Provide both: built and natural provision. Well-maintained playgrounds. Consider sensory qualities like noise and smells.	
We desire fun	The experience of fun	The experience of fun was a central experience for children that were found within all other themes and physical and social environments	

**Table 5 ijerph-20-01763-t005:** Intense play experiences and physical environmental affordances.

Intense Play Experience	Relation to the Physical Environmental Affordances
Great heights, including high swinging, high climbing, elevated balancing, jumping down/up/far/high, high constructions	Play equipment that was larger in size and more elevated was found in swings, crossing bridges, as well as jumping, climbing, and balancing opportunities that elevated [54,61,70,88]. Landscape features that were longer and steeper [64,69,90,96]. Jumping down from high points [59,96] or jumping elevated far distances [92], jumping on built environment structures at great heights [96], or jumping really high on trampolines [84]. Climbing play associated with heights was expressed by children with and without disabilities as climbing as high as possible [57,59,61,72,92,100]. Constructing was related to accessing material that allowed forms to be built as high as possible [92,96]. Kicking a ball high up in the air [66].
Reaching the highest point is a desirable challenge and an opportunity to have an outlook	Play equipment that offered several ways (e.g., climbing net, steps, ramp) to the highest point that is reachable for children regardless of their abilities [56,60,67,76,81,96].
Great speed, including fast swinging and feeling the wind, fast sliding, fast seesawing, fast running, fast zip-lining, fast rolling down, fast driving	Taller, bigger, or longer play equipment or longer, steeper slopes that provided more speed [53,58,61,70,71,78,92,98,100]. Smooth surfaces that allowed fast driving and sliding (e.g., with wheelchairs, scooters, or bikes) [64,72]. A person who is pushing the child fast, spinning, and swinging [53].
Bumpier sliding	Waved slides [56], sledding over bumps [69].
Heavier carrying	Bigger and heavier play objects are transported and moved around [92].
Enjoyment in going slower	Hammocks that allow slow swinging and daydreaming [81,92].
Digging deeper	Material property allowing digging [59,92].
Other environmental aspects affording intense play	Surfaces properties that are uneven, irregular, very smooth, icy, or frosty when children engage in balance, climbing, or slide/gliding [69,90,92,94,96]. Having possibilities for diverse sensory play, including getting messy [84,94], tactile play through touching natural materials [68,76,78] walking barefoot [78,88], tasting herbs, fruits, vegetables, ice, and snow [69,76,78], smelling flowers [78,79], watching spinning objects [53,68], and creating sounds with music instruments [53] or natural elements (crackling dry leaves) [78]. Having possibilities to combine or mix multiple materials, for example, sand play in combination with water, natural loose materials, or manufactured play objects [80,89,90,91,94,101].
Examples of combined intense play experiences	A tall swing that also spins [57]. A slide that additional to great length also incorporates waves to provide a bumpier slide down [56]. A tall swing allowing for swinging fast and jumping off at the right moment [73,88]. A slope with a self-built hump out of snow allows for sliding faster and jumping further [69]. A biking path incorporating a big bump allows for moving up and down quickly [63].
Examples of novel qualities of playgrounds	Newly furnished playgrounds [56,89,92], inclusive play equipment and playgrounds [72,95], natural surroundings and landscape features [58,78,81], themed play equipment e.g., animal-themed [71,77], real climbing trees [78,92,100] and real rocks or boulders [71,89,93], creeks (human constructed) [94] tree houses [81], larger playground with provision of natural loose materials that kept children’s interest for longer periods [54,69,92,94], large obstacles courses [100], provision of play objects and equipment that could be formed and shaped by children [84,94], and weather conditions and seasonal changes providing additional affordances [64,69,76,81,92,96].

## Data Availability

Not applicable.

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
