# Peer review of "Environmental Qualities That Enhance Outdoor Play in Community Playgrounds from the Perspective of Children with and without Disabilities: A Scoping Review"

_ijerph, 2023, doi:10.3390/ijerph20031763_

Round 1

Reviewer 1 Report

Thank you very much for the opportunity to review this article. Overall, I found it an interesting article that may be useful to the scientific community and pediatrics in general. However, some minor aspects could be revised to improve the text. These are the main ones:

- The article could be shorter. I understand that the authors have done exhaustive work in the review performed. However, one of a review's most important tasks is to filter and summarize the most important findings. The introduction could be shortened. The methods section is relatively contained in length. However, the results and discussion sections are excessively long. I recommend reducing both sections as much as possible to make the text easier to read.

- This is a purely aesthetic aspect, not a scientific one, but the colors in Figure 2 and the width of the columns should be revised. The figure is unpleasant to look at as it is.

- In my humble view, table 4 does not make the least sense. I understand that the authors want to reflect on their immense work. However, their most critical work consists of filtering, summarizing, exposing their findings, and not presenting a vast table occupying seven pages of the manuscript. A reader cannot draw any conclusions from that table. I recommend that this table be offered as supplementary material, which readers can review if they feel like it, and that the authors focus on extracting and analyzing the most relevant results from this table. It would make the article easier to read and thus reach many more potential readers.

- There are sections of the results that give the impression that they mix results with discussion, as in 3.2.2, 3.2.3, and the next. I would recommend that the authors revise these sections so that, in the results, they present the findings and place the comparisons or comments in the discussion section. These modifications could also help to reduce the length of both sections.

Author Response

Reviewer comments

Authors answer

Thank you very much for the opportunity to review this article. Overall, I found it an interesting article that may be useful to the scientific community and pediatrics in general. However, some minor aspects could be revised to improve the text. These are the main ones:

Thank you for this overall comment and reviewing the manuscript.

- The article could be shorter. I understand that the authors have done exhaustive work in the review performed. However, one of a review's most important tasks is to filter and summarize the most important findings. The introduction could be shortened. The methods section is relatively contained in length. However, the results and discussion sections are excessively long. I recommend reducing both sections as much as possible to make the text easier to read.

We shortened as much as possible introduction, results, and discussion sections. Some words have been added as we conducted a follow up search as recommended from another reviewer. Changes have been marked yellow.

- This is a purely aesthetic aspect, not a scientific one, but the colors in Figure 2 and the width of the columns should be revised. The figure is unpleasant to look at as it is.

Thank you for this comment. We adjusted width of the columns to be greater and we adjusted colours following recommendations of Universal Design to accommodate colour blindness and colour vision deficiency. Additionally, we adjusted size of the font in figure 2.

- In my humble view, table 4 does not make the least sense. I understand that the authors want to reflect on their immense work. However, their most critical work consists of filtering, summarizing, exposing their findings, and not presenting a vast table occupying seven pages of the manuscript. A reader cannot draw any conclusions from that table. I recommend that this table be offered as supplementary material, which readers can review if they feel like it, and that the authors focus on extracting and analysing the most relevant results from this table. It would make the article easier to read and thus reach many more potential readers.

We agree with your comment on the very lengthy table, which is providing more an overview of the included studies. We included this table as this is a common approach for scoping reviews. However, we agree that this table could be reviewed if the reader is interested. We moved it in the appendix, renamed it Appendix B and provided information in the main text that allows reader to locate the table in the appendix if needed.

We adjusted all table numbering throughout the manuscript.

- There are sections of the results that give the impression that they mix results with discussion, as in 3.2.2, 3.2.3, and the next. I would recommend that the authors revise these sections so that, in the results, they present the findings and place the comparisons or comments in the discussion section. These modifications could also help to reduce the length of both sections.

We reviewed the result section and made changes. We also paid attention in writing style and grammar in these sections as we realized this might have been switched from past to present tense. Partly we deleted sections or sentences that are elaborated on in the discussion. We are confident these sections are more comprehensive now.

Reviewer 2 Report

The article is well done. I appreciate the images that show the data specifics. I do question the search terms as I wonder if you may have inadvertently missed some studies. Missing are terms like "school yard," "recess," "break time,"  "park," "equipment." I wished there had been some reflection on the perceived utility of your own search terms in the discussion section.  I do love the language of being an "adult ally," although it could be even clearer, as in "an adult ally of children." 

Author Response

Reviewer comments

Author answer

The article is well done. I appreciate the images that show the data specifics. I do question the search terms as I wonder if you may have inadvertently missed some studies. Missing are terms like "school yard," "recess," "break time,"  "park," "equipment." I wished there had been some reflection on the perceived utility of your own search terms in the discussion section.  

Thank you for your comment on potential missed search terms. Some of these potential keywords that you suggested were documented and used for test searches but were found not to be included in the final search string. Our aim was to use search terms broad to the topic allowing us to include different community contexts such as schools and parks.

I do love the language of being an "adult ally," although it could be even clearer, as in "an adult ally of children." 

Thank you for this suggestion. We agree with your suggestion and have made changes in the manuscript accordingly to your comment.

Reviewer 3 Report

Title: Environmental Qualities that Enhance Outdoor Play on Community Playgrounds from the Perspective of Children with and 3 without Disabilities: A Scoping Review

Generally, paper is well structured, important theoretical aspects of the examined problem are studied and presented in a clear and consistent manner. Literature is relevant. Research gap is clearly indicated.  According reviewer best knowledge the proposed approach is new. Generally speaking, the paper is well-structured and research gap is clearly indicated.  In my opinion the subject of the paper is directly related to the Journal's main topics.

1. The abstract does not look impressive. An abstract should address these concerns: what are you trying to do, why, what you found and what is the significance of your findings. Rewrite and improve.

2. A discussion on the contribution of the paper is needed. What are the advantages and disadvantages of the proposed method?

3. Please add one paragraph at the end of introduction section to describe the whole structure or process of the manuscript.

4. The novelty and contribution are missing in the paper. Please properly describe in the introduction section.

5. The writing is recommended to be improved.

6. Findings, limitations, and recommendations of this paper can be discussed more in the conclusion section.

Author Response

Reviewer comments

Author answer

Title: Environmental Qualities that Enhance Outdoor Play on Community Playgrounds from the Perspective of Children with and 3 without Disabilities: A Scoping Review

Generally, paper is well structured, important theoretical aspects of the examined problem are studied and presented in a clear and consistent manner. Literature is relevant. Research gap is clearly indicated.  According reviewer best knowledge the proposed approach is new. Generally speaking, the paper is well-structured and research gap is clearly indicated.  In my opinion the subject of the paper is directly related to the Journal's main topics.

Thank you very much for this comment. We understand that scoping reviews as a systematic literature review might be not a common method depending on the discipline. Your comments are much appreciated as those provide a new perspective to improve the manuscripts readability for potential reader who might not be familiar with scoping reviews.

1. The abstract does not look impressive. An abstract should address these concerns: what are you trying to do, why, what you found and what is the significance of your findings. Rewrite and improve.

Thank you for your comment and guidance to rewrite the abstract. We have restructured the abstract what now includes the aim, rational why this study is important, we provided more details in methods (such as when last searched, disciplines included) and rewrote findings, discussion topics and conclusions.

2. A discussion on the contribution of the paper is needed. What are the advantages and disadvantages of the proposed method?

We rephrased the first paragraph in the method section. In this paragraph we defined what a scoping review is and its advantages. In the limitations we discussed disadvantages.

3. Please add one paragraph at the end of introduction section to describe the whole structure or process of the manuscript.

Thank you for this comment. We understand that your suggestion on adding a paragraph in the introduction that describe the structure and process of the manuscript should increase readability. However, such an approach is not typical for a scoping review following the methodological and reporting guidelines that this scoping review  followed from Arksey and O’Malley [1], Levac et al. [2] and guidelines from the Joanna Briggs Institute [3] as well as other already published scoping reviews. In addition, we did not find such a recommendation from the author guidelines from the Journal. Therefore, we anticipate such a paragraph might confuse potential reader as they will not find this in other published scoping reviews and papers from this journal. We hope that following these guidelines and common approaches in scoping reviews (such as how the method section and results are structured) provide the structure to increase readability.

We formulated a short paragraph and if you still think we should add it into the manuscript we can add it:

This scoping review is structured by first by an introduction that defines important concepts and describes research that has already been done in topic of interest. Second, the method section describes the five steps of conducting a scoping review following guidelines from Arksey and O’Malley [1], Levac et al. [2] and guidelines from the Joanna Briggs Institute [3]. The results present the study scope included and results of a qualitative content analysis. The discussion elaborates on the main findings, highlights the identified gaps and a reflection on limitations and further research is presented.

4. The novelty and contribution are missing in the paper. Please properly describe in the introduction section.

We agree with your comment, and we tried to provide more information on already published reviews on similar topics to highlight the novelty of this review. We put this information in the last paragraph of the introduction and link it with the aim of this scoping review. We hope this is clearer and provide insight in the significance of this study.

5. The writing is recommended to be improved.

A co-author who is a native speaker prove read the whole manuscript.

6. Findings, limitations, and recommendations of this paper can be discussed more in the conclusion section.

We rewrote the conclusion added following details in the conclusion accordingly to your suggestions: -

-        rewrote the findings

-        added main gaps identified. addressed the limitations in separate heading above the conclusion.

Reviewer 4 Report

The revised manuscript addresses an issue of great importance to ensure proper child development. Specifically, it reviews the published scientific literature on the vision of children with and without disabilities regarding the characteristics that playgrounds should have. Play is a fundamental way of learning and socialization for children and for human beings in general, although this aspect is very often undervalued in educational centers. Another of the strengths of this research lies in focusing on the voices of the protagonists of the playgrounds themselves, —the children—, who can provide valuable insights for their planning, design and evaluation. It should not be ignored that the study also considers the opinions of children with disabilities, a group very often marginalized in this type of research. For all these reasons, I would like to offer my sincere congratulations to the authors of the manuscript.

However, in order to contribute to the improvement of the study, I propose to review some aspects:

·     - Include figures and tables closer to the point where they are cited in the text. It is a large manuscript with a high number of graphic elements, so this better location of the figures and tables would facilitate reading and understanding of the information.

    -The literature review is extensive and diverse. However, only 49.14% of the sources consulted date from the last 5 years. It is therefore suggested that some references be updated.

     -The review was carried out in 7 databases: “relevants to disciplines of health, education, social sciences, and architecture, which allowed a broad range of literature from different professional audiences to be included” (p. 3). However, it stands out to me the fact that ERIC was not considered. What are the reasons for its exclusion?

   -It is not clear whether the coding process was triangulated: “Analysis was performed by the first author (T.M.) and guided by group discussions with the review team” (p. 5). If so, in what form was a concordance test performed?

   -In the review carried out, were any differences identified in any article between the views of boys and girls? With a view to future studies, it would be interesting to investigate how friendly play spaces are for girls and whether they feel, in any way, discriminated against or excluded.

Author Response

Reviewer comments

Author answer

The revised manuscript addresses an issue of great importance to ensure proper child development. Specifically, it reviews the published scientific literature on the vision of children with and without disabilities regarding the characteristics that playgrounds should have. Play is a fundamental way of learning and socialization for children and for human beings in general, although this aspect is very often undervalued in educational centers. Another of the strengths of this research lies in focusing on the voices of the protagonists of the playgrounds themselves, —the children—, who can provide valuable insights for their planning, design and evaluation. It should not be ignored that the study also considers the opinions of children with disabilities, a group very often marginalized in this type of research. For all these reasons, I would like to offer my sincere congratulations to the authors of the manuscript. However, in order to contribute to the improvement of the study, I propose to review some aspects:

Thank you very much for this overall comment and your endorsement to this manuscript.

    - Include figures and tables closer to the point where they are cited in the text. It is a large manuscript with a high number of graphic elements, so this better location of the figures and tables would facilitate reading and understanding of the information.

We adjusted the location of the tables and figures. Some adjustments needed to be made to not split tables over two pages. For example, we wanted that table 4 is readable and presented on one page not over two pages as they present important results.

    -The literature review is extensive and diverse. However, only 49.14% of the sources consulted date from the last 5 years. It is therefore suggested that some references be updated.

Thank you for this comment and we agree and updated a search. Six new studies were included. The review process followed the same procedure as in the first search. This has not significantly changed the results. Everything that was added new is presented in yellow.

     -The review was carried out in 7 databases: “relevants to disciplines of health, education, social sciences, and architecture, which allowed a broad range of literature from different professional audiences to be included” (p. 3). However, it stands out to me the fact that ERIC was not considered. What are the reasons for its exclusion?

Thank you for this comment. During the process of selecting the databases we tried to consider databases from different disciplines. We considered ERIC as a potential database during preparation of the scoping review. Test searches were conducted in ERIC but did not show relevant hits. After consultation with information specialists from the university library we decided on other databases that included both school and public contexts. This is also represented in that studies from educational field are represented with 15 research papers.

   -It is not clear whether the coding process was triangulated: “Analysis was performed by the first author (T.M.) and guided by group discussions with the review team” (p. 5). If so, in what form was a concordance test performed?

We agree it was not clear enough how we ensured trustworthiness in the findings.

We added more details for clarity: “Analysis started with a joint coding of the first four studies (approximate 8% of all studies) with the review team. The other studies were analysed by the first author (T.M.) and guided by discussions with the whole review team to confirm formed codes and themes. The authors of this review are experienced occupational therapists, and two of them (C.S., H.L.) have advanced knowledge of research on children’s play.”

   -In the review carried out, were any differences identified in any article between the views of boys and girls? With a view to future studies, it would be interesting to investigate how friendly play spaces are for girls and whether they feel, in any way, discriminated against or excluded.

Thank you for this comment. You address an important and contemporary topic. This scoping review was not aiming specifically investigating the difference between perspectives of boys and girls. Therefore, we did not analysed studies specifically regarding boys’ and girls’ differences. We belief that an analysis on sex differences is valuable but would extent the aim of this research and need to be addressed in an own review.
